# BlobCUT: A Contrastive Learning Method to Support Small Blob Detection in Medical Imaging

**DOI:** 10.3390/bioengineering10121372

**Published:** 2023-11-29

**Authors:** Teng Li, Yanzhe Xu, Teresa Wu, Jennifer R. Charlton, Kevin M. Bennett, Firas Al-Hindawi

**Affiliations:** 1School of Computing and Augmented Intelligence, Arizona State University, Tempe, AZ 85281, USA; tengli3@asu.edu (T.L.); yanzhexu@asu.edu (Y.X.); falhinda@asu.edu (F.A.-H.); 2Division Nephrology, Department of Pediatrics, University of Virginia, Charlottesville, VA 22903, USA; jrc6n@hscmail.mcc.virginia.edu; 3Department of Radiology, Washington University, St. Louis, MO 63130, USA; kmbennett@wustl.edu

**Keywords:** imaging biomarker, blob detection, glomeruli segmentation, Hessian analysis, contrastive learning

## Abstract

Medical imaging-based biomarkers derived from small objects (e.g., cell nuclei) play a crucial role in medical applications. However, detecting and segmenting small objects (a.k.a. blobs) remains a challenging task. In this research, we propose a novel 3D small blob detector called BlobCUT. BlobCUT is an unpaired image-to-image (I2I) translation model that falls under the Contrastive Unpaired Translation paradigm. It employs a blob synthesis module to generate synthetic 3D blobs with corresponding masks. This is incorporated into the iterative model training as the ground truth. The I2I translation process is designed with two constraints: (1) a convexity consistency constraint that relies on Hessian analysis to preserve the geometric properties and (2) an intensity distribution consistency constraint based on Kullback-Leibler divergence to preserve the intensity distribution of blobs. BlobCUT learns the inherent noise distribution from the target noisy blob images and performs image translation from the noisy domain to the clean domain, effectively functioning as a denoising process to support blob identification. To validate the performance of BlobCUT, we evaluate it on a 3D simulated dataset of blobs and a 3D MRI dataset of mouse kidneys. We conduct a comparative analysis involving six state-of-the-art methods. Our findings reveal that BlobCUT exhibits superior performance and training efficiency, utilizing only 56.6% of the training time required by the state-of-the-art BlobDetGAN. This underscores the effectiveness of BlobCUT in accurately segmenting small blobs while achieving notable gains in training efficiency.

## 1. Introduction

Imaging biomarkers are measurable features or characteristics that are detected using medical imaging technologies, such as magnetic resonance imaging (MRI), computed tomography (CT), positron emission tomography (PET), or ultrasound. These biomarkers provide objective and quantitative measures of disease or injury and are often used in medical research and clinical practice to diagnose, monitor, and evaluate the progression or treatment response of the disease. In particular, some important imaging biomarkers are derived from small, densely distributed structures (e.g., cell nuclei, kidney glomeruli) known as “blobs” in vivo. The characterization of blobs based on features such as shape, size, and quantity has led to their use as imaging biomarkers to aid in diagnosis, prognosis, and treatment planning [1,2,3,4]. Thus, the detection and segmentation of blobs in medical images are critical for accurately assessing and interpreting disease states.

Similar to its versatile applications in various fields like manufacturing [5,6] and pool boiling [7,8,9], Artificial Intelligence (AI) has emerged as a transformative force in the field of medicine as well, offering numerous advantages that enhance patient care and streamline healthcare processes. One of the key benefits is the ability of AI to analyze vast amounts of medical data quickly and accurately, aiding in early disease detection and diagnosis. Machine learning algorithms can identify patterns in medical images, such as X-rays and MRIs, enabling healthcare professionals to make more informed decisions [10,11]. Concurrently, the escalating demand for AI support in the medical domain serves as a catalyst for advancements in various realms of computer science, notably in the domain of computer vision.

However, detecting and segmenting blobs is a difficult computer vision task for several reasons. Firstly, acquiring labeled data, a prerequisite for numerous deep learning algorithms, which is costly for blob-like data, proves to be an exceedingly challenging task. And blobs have varying and often irregular shapes, which complicates the segmentation process. Additionally, medical images are often characterized by low resolution, image noise, and other challenges that hinder the accurate segmentation of small, intricate blobs. Moreover, the detection and recognition of blobs’ boundaries are challenging, especially when blobs overlap. As a result, sophisticated algorithms are necessary to address these challenges and accurately detect and segment blobs in medical images.

Current methods of segmentation often struggle to handle the intricacies of blob detection, particularly in medical imaging. Traditional approaches, such as U-Net [12] and U-Net-like models [13,14], represent a class of fully supervised models commonly employed in various image segmentation tasks. However, their efficacy is contingent upon the availability of annotated data, a prerequisite that becomes particularly challenging and costly, especially in scenarios involving the detection of small blobs. Although leveraging the advantages of transfer learning techniques, which aim to capitalize on knowledge gained from related tasks, can mitigate the problem, it may face challenges in adapting to domain shifts, and the pre-trained knowledge may not always be perfectly aligned with the target task, potentially leading to suboptimal performance in certain scenarios. Moreover, U-Net and U-Net-like models often depend on thresholding techniques applied to probability maps. The problem with this method is that under-segmentation arises when small blobs or objects are not properly segmented due to the thresholding process. When segmenting small blobs, setting a fixed threshold may lead to the exclusion of parts of the object that fall below the threshold, even if they are part of the object of interest. This can result in incomplete segmentation and missing small but crucial details, leading to a loss of information and accuracy in the segmentation task, as demonstrated by Xu et al. [14].

The importance of efficient small blobs segmentation techniques holds paramount importance in the realm of medical imaging and diagnostics. Specifically, it is evident in medical applications such as identifying glomeruli from kidney Cationic-ferritin-enhanced MR (CFE-MR) images which are roughly spherical and have blob-like shapes in 3D space. Glomeruli, being the functional units of the kidney, play a crucial role in the filtration of blood and the formation of urine. Accurate identification and segmentation of glomeruli in three-dimensional space contribute to a better understanding of renal function, aiding in the diagnosis and monitoring of kidney-related disorders. The 3D nature of glomeruli, with their roughly spherical and blob-like shapes, poses a unique challenge for segmentation techniques. The ability to precisely detect and segment these structures can provide valuable insights into the health of the kidneys.

In light of the limitations in current segmentation methods and the impact of addressing these limitations on the medical imaging and diagnosis field, the proposed work aims to overcome these challenges by introducing a novel approach that allows unpaired image-to-image translation from medical images to label masks. The proposed model, named BlobCUT, is motivated by the CUT model [15] and specifically designed for the detection and segmentation of glomeruli. This innovative method involves a multi-step process to achieve robust segmentation. First, a 3D elliptical Gaussian function is utilized to synthesize 3D blobs and their masks, simulating the geometric properties of glomeruli. This generated data serves as the clean data domain (target domain) for training, to which the noisy glomeruli data needs to be translated. Second, a modification to the CUT model is introduced, incorporating a convexity consistency constraint to minimize the impact of noise and ensure the preservation of blobs’ locations, shapes, and data distributions. Third, a distribution consistency constraint based on Kullback–Leibler divergence is proposed to maintain the same distribution between denoised and synthetic blob images, leveraging information at the voxel level. Finally, the model is capable of denoising glomerular images and deriving the Hessian convexity mask to facilitate blob identification.

To validate the performance of BlobCUT, two studies were conducted. The first study is conducted on a 3D simulated blobs dataset where the locations of blobs are known. The study included six methods from the literature for comparasion, namely UH-DoG [13], BTCAS [14], CUT [15], UVCGAN [16], EGSDE [17], and the state-of-the-art BlobDetGAN [18]. The second study was performed on a set of 3D images of mouse kidneys from CFE-MRI, where BlobCUT was compared against UH-DoG, BTCAS, UVCGAN, EGSDE, BlobDetGAN, and stereology [19]. This comprehensive assessment aims to demonstrate the efficacy and superiority of BlobCUT in addressing the challenges posed by current segmentation methods in the context of glomeruli identification. In this context, the proposed 3D BlobCUT model becomes not only a technical innovation but also a vital tool for advancing our understanding of renal physiology and pathology, making it a crucial application in the field of medical image segmentation.

In summary, this research presents three significant contributions:A novel small blob detector: the proposed BlobCUT leverages Generative Adversarial Network (GAN) and Contrastive Learning to address the challenge of limited labeled data. By incorporating the assumption of geometric properties, BlobCUT offers a novel, effective and efficient solution for detecting small blobs.Novel constraints for improved performance: This work introduces a novel convexity consistency constraint, based on Hessian analysis, and a unique blob distribution consistency constraint, based on Kullback–Leibler divergence. These constraints are designed to preserve the geometry property and intensity distribution of blobs, resulting in enhanced segmentation performance.Comprehensive performance evaluation: Extensive comparisons were conducted with six state-of-the-art methods, spanning both simulated and real MRI datasets. The outcomes of these comparisons affirm the superior performance of BlobCUT, underlining its effectiveness in small blob detection compared to existing methodologies.

The rest of the paper is organized as follows. Section 2 illustrates the advantages and disadvantages of some related works. Section 3 presents our methodology to detect and segment small blobs. Section 4 describes our experiment setup and results evaluation. Section 5 discusses the potential to apply the proposed method to another medical scenario and other areas. Section 6 concludes contributions and results followed by discussing limitations and future work.

## 2. Literature Review

Several works have been introduced to address the small blob detection problem. Kong et al. proposed a generalized Laplacian of Gaussian (gLoG) [20] filter for detecting general elliptical blob structures in images. The gLoG detector is able to accurately locate the blob centroids and estimate the sizes, shapes, and orientations of the detected blobs. Zhang et al. developed a series of blob detectors including the Hessian-based Laplacian of Gaussian (HLoG) [21] and the Hessian-based Difference of Gaussian (HDoG) [22]. The advantage of the Hessian analysis is that it can automatically extract the regional features, but the problem is that it makes the unsupervised clustering-based post-pruning more sensitive to noise than the traditional threshold-based post-pruning [23] commonly used in most imaging detectors [13]. Moreover, blobs’ background in medical images is often heterogeneous (e.g., uneven light of background), leading to the over-detection problem [14].

Recent studies have demonstrated the effectiveness of deep-learning-based segmentation methods in detecting and segmenting large objects in medical images [1,3,4,24,25]. However, due to the small size and dense distribution of blobs in the images, the performance of such methods on blob segmentation is questionable [13]. Thresholding on the probability map from commonly used deep learning methods, such as U-net, has been shown to lead to under-segmentation problems when segmenting small blobs, as demonstrated by Xu et al. [14]. Instead, researchers have explored using deep learning models’ denoising ability to support blob detection. U-Net-based methods have been proposed for blob detection, such as the UH-DoG blob detector [13], which was developed primarily for detecting glomeruli in Cationic-ferritin-enhanced MRI (CFE-MRI) images. The UH-DoG model was pre-trained on a nuclei dataset, and the blobs’ probability map in the CFE-MR images was derived. It was then combined with a Hessian convexity map to identify true glomeruli. To improve the computational efficiency and segmentation accuracy of glomeruli detection and segmentation, BTCAS [14] was developed as an extension of the UH-DoG detector. This approach transforms blobs to local optimum DoG space adaptively by utilizing the bounded scales from U-Net.

Essentially, methods based on U-Net are supervised and heavily dependent on the quality of the training dataset. However, the scarcity and costliness of labeled medical data impose constraints on the applicability of U-Net-based approaches for blob segmentation. To overcome the issues of noisy medical data and the rarity of labels, researchers started utilizing unpaired image-to-image (UI2I) translation models which are label-free. UI2I translation models are machine learning algorithms used to transform images from one domain to another without the need for paired data. One popular type of unpaired image-to-image translation model is based on generative adversarial network (GAN) [26], which consists of two neural networks: a generator and a discriminator. The generator network learns to create new images similar to the target domain, while the discriminator network learns to distinguish between the generated images and real images from the target domain. The two networks are trained together in a game-like process, where the generator tries to create more realistic images and the discriminator tries to identify the generated images. This process continues until the generator is able to create images that are indistinguishable from real images in the target domain. These models can be used for a variety of tasks, such as style transfer [27], image colorization [28], and image super-resolution [29]. Utilizing the power of GAN, Zhao et al. [17] proposed EGSDE which uses energy-guided stochastic differential equations to help training, and Torbunov et al. [16] further integrated a CycleGAN [30] with U-Net-ViT [31], achieving state-of-the-art performance in UI2I tasks.

Recently, researchers started realizing the potential of UI2I translation to solve image detection and segmentation problems. By considering objects with noise as one domain and objects without noise as another domain, the noisy data is translated into the clean data’s domain without the need for any labels. Xu et al. [32] proposed BlobDetGAN, a CycleGAN-based model to solve blob detection in two steps: image denoising and object segmentation. BlobDetGAN is entirely label-free as it leverages both the blobs’ geometric properties and noise distribution to denoise the real noisy image to the clean image with only target blobs. However, there are two disadvantages to using CycleGAN in this application. The first is that CycleGAN consists of two generators and two discriminators and was designed to support two-way direction training (Domain A → Domain B and Domain B → Domain A). However, training four networks simultaneously makes it unnecessary and time-consuming since only one direction is required for the denoising application. The second disadvantage is that the cycle consistency constraint used in CycleGAN-based models is sometimes too restrictive to the relationship between the two domains [15]. Taesung et al. simplified the CycleGAN from two generators and two discriminators to only one generator and one discriminator by using contrastive learning to propose a Contrastive Unpaired Translation (CUT) model [15]. CUT offers an alternative straightforward way of maintaining correspondence in content but not appearance by maximizing the mutual information between the corresponding input and output images. Although CUT may have better image translation capability and help with reducing the time required for training, it will not work well directly when denoising images that consist of small objects such as blobs and background noise. The aforementioned statement is attributed to the fact that CUT utilizes the dissimilarities between image patches as a source of information. Notably, these patches are not specifically engineered to detect discrepancies at the level of individual pixels/voxels. Rather, the focus of the method is on extracting collective statistics from the comparison of patches. In our denoising problem, pixels/voxels are also as important as the collective statistics because small blobs have similar sizes, shapes, and data distributions with noise and a pixel/voxel change may cause big differences in biomarker extraction. Thus, the CUT model needs to be modified to work properly to detect small blobs.

## 3. Materials and Methods

As Figure 1 illustrates, our proposed BlobCUT consists of three steps to identify the blobs (e.g., glomeruli) from the kidney CFE-MRI images: (1) Synthesize 3D blobs by using the 3D elliptical Gaussian function and randomly generate 3D blobs image with their blob mask as training input and shape constraints; (2) train a 3D CUT model with convexity consistency constraint and distribution constraint to translate real 3D blob images (e.g., real kidney CFE-MRI) to clean 3D image and produce a probability mask and a Hessian convexity mask; (3) A voxel union constraint operation is applied on the probability mask and the Hessian convexity mask to derive the final identification mask of blobs.

### 3.1. 3D Blob Synthesis through 3D Gaussian Function

Our proposed BlobCUT includes blob synthesis as the first step of blob detection. Blobs can range in size, shape, orientation and location in images. Some work has been done to use Gaussian function to construct small blobs in 2D space [20,33]. However, in 3D space, the shape and orientation of blobs are more complex. BlobDetGAN [32,34] firstly simulates 3D blobs by using the 3D elliptical Gaussian function from Equation (Equation 1), and it is proven to be effective and efficient in the literature.
(1)$$F\left(x,y,z\right)=A\cdot e^{-(ax^{2}+by^{2}+cz^{2}+dxy+eyz+fxz)}$$
(2)$$\begin{array}{cc} a & =\frac{\sin^{2}\theta \cos^{2}\varphi }{\sigma_{x}^{2}}+\frac{\sin^{2}\theta \sin^{2}\varphi }{\sigma_{y}^{2}}+\frac{\cos^{2}\theta }{\sigma_{z}^{2}} \end{array}$$
(3)$$\begin{array}{cc} b & =\frac{\cos^{2}\theta \cos^{2}\varphi }{\sigma_{x}^{2}}+\frac{\cos^{2}\theta \sin^{2}\varphi }{\sigma_{y}^{2}}+\frac{\sin^{2}\theta }{\sigma_{z}^{2}} \end{array}$$
(4)$$\begin{array}{cc} c & =\frac{\sin^{2}\varphi }{\sigma_{x}^{2}}+\frac{\cos^{2}\varphi }{\sigma_{y}^{2}} \end{array}$$
(5)$$\begin{array}{cc} d & =\frac{\sin2\theta \cos^{2}\varphi }{\sigma_{x}^{2}}+\frac{\sin2\theta \sin^{2}\varphi }{\sigma_{y}^{2}}-\frac{\sin2\theta }{\sigma_{z}^{2}} \end{array}$$
(6)$$\begin{array}{cc} e & =-\frac{\cos\theta \sin2\varphi }{\sigma_{x}^{2}}+\frac{\cos\theta \sin2\varphi }{\sigma_{y}^{2}} \end{array}$$
(7)$$\begin{array}{cc} f & =-\frac{\sin\theta \sin2\varphi }{\sigma_{x}^{2}}+\frac{\sin\theta \sin2\varphi }{\sigma_{y}^{2}} \end{array}$$
where *x*, *y* and *z* are the coordinates of voxels near the blob center which is fixed at the origin point. The shape and orientation of 3D blobs are controlled by coefficients *a*, *b*, *c*, *d*, *e* and *f* via θ, φ, $\sigma_{x}$, $\sigma_{y}$ and $\sigma_{z}$. θ and φ are two angles in the spherical coordinate system, and σ defines the length along each direction. A is a normalization factor. More details are illustrated in Appendix A.

In our study, we employ the aforementioned formulas to generate 3D blobs in the same way of [32,34] and designate their masks as ground truth. During the synthesis of these images, the parameters, location, and quantity of blobs within each image are chosen randomly. Our primary focus is on segmenting glomeruli from kidneys, and therefore, BlobCUT is used to convert real 3D blob images (glomeruli images) into approximate synthesized blob images. The main purpose of synthesizing these 3D blob images is to approximate the actual distribution of the target blobs (glomeruli) while retaining their information, and only eliminating contextual details during the image translation process.

### 3.2. 3D Blob Images Detecting through 3D GAN with Contrastive Learning

We wish to translate images from the input domain–real noisy 3D glomeruli which we denote as the source domain *S*, to appear like an image from the output domain–synthetic clean 3D blob which we denote as the target domain *T*. We are given a dataset of unpaired instances. Our method only requires learning the mapping in one direction and avoids using inverse auxiliary generators and discriminators which are used in CycleGAN-based model. This can largely simplify the training procedure and reduce training time.

With training samples of real noisy blob images $\{I_{n}\}\in S$ and clean synthetic blob images $\{I_{c}\}\in T$, we denote their respective data distributions as $I_{n}∼p_{data}\left(I_{n}\right)$ and $I_{c}∼p_{data}\left(I_{c}\right)$.

The generator *G* consists of two parts, an encoder network $G_{enc}$ and a decoder network $G_{dec}$, which are applied sequentially to produce output image $y=G\left(z\right)=G_{dec}\left(G_{enc}\left(x\right)\right)$. To facilitate effective model training, multiple loss functions were incorporated. The first is the **Adversarial loss**. Adversarial loss function [26] is a common loss function used in GAN training to incentivize the generated output to exhibit visual congruity with images originating from the designated target domain, as follows:(8)$$\begin{array}{cc} L_{GAN}\left(G,D\right)= & E_{I_{c}∼p_{data}\left(I_{c}\right)}\left[\log\left(D\left(I_{c}\right)\right)\right]+\\ \; & E_{I_{n}∼p_{data}\left(I_{n}\right)}\left[1-\log\left(D\left(G\left(I_{n}\right)\right)\right)\right] \end{array}$$
where *G* and *D* are the functions of the generator and discriminator of the basic GAN model of CUT, *X* is the input image, which is typically a real 3D glomeruli image, and *Y* is the output image, which is denoised 3D glomeruli image in our situation.

Another loss function that was incorporated is a loss function based on the **Contrastive learning loss.** The overall target of unpaired image-to-image translation methods is to associate the input and output data, and the loss function is the criterion to measure the correlations of images from different domains. However, traditional unpaired image-to-image translation methods suffer from the same issue-the need for a pre-specified, hand-designed loss function to measure predictive performance [35,36]. Recently, a family of methods based on maximizing mutual information has emerged to bypass the above issue [35,36,37,38,39,40,41]. These approaches employ noise contrastive estimation (NCE) [42] for the purpose of learning an embedding space that brings related signals into proximity while distinguishing them from other samples within the dataset. It is noteworthy that analogous concepts have historical roots in seminal work on metric learning, exemplified by Siamese networks [43]. Associated signals can be an image with itself [38,41,44,45,46], an image with its downstream representation [47], neighboring patches within an image [35,39,48], or multiple views of the input image [49], and most successfully, an image with a set of transformed versions of itself [37,40]. Thus, by maximizing the mutual information, we can address the above issue.

Contrastive Unpaired Translation (CUT) model first uses patchwise contrastive loss [35] under a noise contrastive estimation (NCE) framework [35] to maximize the mutual information between input and output at the patch level. It can enable one-sided image translation, which is more efficient than CycleGAN-based methods, with comparable good quality. The core concept underlying contrastive learning is the establishment of associations between two signals, denoted as the ’query’ and its corresponding ’positive’ example, as opposed to the remaining data points within the dataset, which are commonly referred to as ’negatives’. Thus, the cross-entropy loss can be calculated, representing the probability of the positive example being selected over the negatives. In our context, query refers to a small patch of clean 3D glomeruli images (output). Positives and negatives are the same and different position patches from real noisy 3D glomeruli images (input). For example, given a patch showing a lot of glomeruli from the output image, one should be able to more strongly associate it with the corresponding cortical area from the input image, more so than the other patches of the medullar areas or other noise areas. Even at the pixel/voxel level, the intensity of a glomerulus (close to zero) can be more strongly associated with the intensity of the glomerulus than with the background noise. Thus, we employ a multilayer, patch-based learning objective. We aim to match corresponding input-output patches at a specific location area (cortex, medullar, and noisy area). We can leverage the other area patches within the input as negatives. The multilayer patchwise contrastive loss (denoted as PatchNCE loss), as illustrated in Figure 2, is selected to be used:(9)$$L_{PatchNCE}\left(G,F,X\right)=E_{x∼X}\sum_{l=1}^{L}\sum_{s=1}^{S_{t}}l\left(z_{l}^{s},z_{l}^{S/s}\right)$$
where *x* is a patch from an image *X*, and *F* is the feature map generated from a two-layer fully connected network after the generator. We pass patches of the image through a small two-layer MLP network *F* same as [37], generating features $z_{l}^{s}=F_{l}\left(G_{enc}\left(x\right)\right)$ and the other features in other locations as $z_{l}^{S/s}$. *l* is the layer index l∈1,2,⋯,L and $s\in 1,2,\cdots ,S_{l}$ where $S_{l}$ is the number of locations in images. You can find more details in [15]. In our works, we pass both source domain images $I_{n}$ and target domain images $I_{c}$ into the network to guarantee the translation quality. Thus we have two patch NCE loss $L_{PatchNCE}\left(G,F,I_{n}\right)$ and $L_{PatchNCE}\left(G,F,I_{c}\right)$.

However, based on the above settings, any small blobs’ geometric transformation in the translation have little effect on their distribution and cannot be identified by discriminators. The CUT utilizes the dissimilarities between image patches as a source of information. However, it is important to note that these patches are not specifically engineered to detect discrepancies at the level of individual pixels/voxels. Rather, the focus of the method is on extracting collective statistics from the comparison of patches. In our denoising problem, pixels/voxels are also as important as the collective statistics because small blobs have similar sizes, shapes, and data distributions with noise and a pixel/voxel change may cause big differences in biomarker extraction. Normally, directly applying the CUT model to kidney MRI images will cause geometric distortion and object blur, which essentially manifest as intensity changes at the pixel/voxel level. Thus, the translated 3D blobs in $G_{dec}$ with the geometric distortion will have an inconsistent geometric property with the glomeruli in $Real_{A}$. Therefore, in addition to the above losses, we propose a **Convexity consistency constraint loss** that is based on Hessian analysis in BlobCUT to preserve blob voxels’ convexity invariance to satisfy the convexity property of glomeruli.
(10)$$\begin{array}{cc} L_{convex}\left(G,HI,M\right)= & E_{I_{n}∼p_{data}\left(I_{n}\right)}[||(HI\left(J-G\left(I_{n}\right)\right)\\ \; & -HI\left(J-I_{n}\right))⊙M\left(I_{n}\right){\left|\right|}_{1}] \end{array}$$
where *J* denotes the unit matrix with all ones, and HI(f) is the Hessian indicator to measure the convexity of voxels in *f*, and it is defined by the corresponding Hessian matrix H(·). The details can be found at Equations (Equation 11) and (Equation 12)
(11)$$H\left(I_{f}\left(i,j,k\right)\right)=\left[\begin{array}{ccc} \frac{\partial^{2}I_{f}\left(i,j,k\right)}{\partial i^{2}} & \frac{\partial^{2}I_{f}\left(i,j,k\right)}{\partial i\partial j} & \frac{\partial^{2}I_{f}\left(i,j,k\right)}{\partial i\partial k}\\ \frac{\partial^{2}I_{f}\left(i,j,k\right)}{\partial i\partial j} & \frac{\partial^{2}I_{f}\left(i,j,k\right)}{\partial j^{2}} & \frac{\partial^{2}I_{f}\left(i,j,k\right)}{\partial j\partial k}\\ \frac{\partial^{2}I_{f}\left(i,j,k\right)}{\partial i\partial k} & \frac{\partial^{2}I_{f}\left(i,j,k\right)}{\partial j\partial k} & \frac{\partial^{2}I_{f}\left(i,j,k\right)}{\partial k^{2}} \end{array}\right]$$
(12)$$HI_{\left(i,j,k\right)}\left(I_{f}\right)=\left\{\begin{array}{cc} 1 & H(I_{f}\left(i,j,k\right))<0\\ 0 & H(I_{f}\left(i,j,k\right))\geq 0 \end{array}\right.$$

This binary indicator matrix (referred subsequently to as Hessian convexity mask) identifies the blob voxels as 1 and the rest as 0. To preserve the geometry of 3D blobs in the translated images $G(I_{n})$, $G(I_{n})$ should have fewer convex voxels than $I_{n}$. Since $G(I_{n})$ is a normalized image with darker blobs as compared to the background, we define the convexity consistent constraint as:(13)$$\sum_{v=1}^{N}HI_{v}\left(J-G\left(I_{n}\right)\right)\leq \sum_{v=1}^{N}HI_{v}\left(J-I_{n}\right)$$
where *J* denotes the unit matrix with all ones, *N* denotes the total number of voxels in $I_{n}$, and *v* denotes index of voxel in $I_{n}$. Equation (Equation 13) means the translated images should always have less or equal total value of Hessian indicator (i.e., less or equal convex objects) than that of input images.

Finally, It is not enough for the generative model to have the convexity consistency constraint to denoise the real glomeruli, because the original CUT was not designed to capture pixel/voxel differences. Thus, we enforce a **Probability distribution constraint** which involves a Kullback-Leibler divergence (KLD) loss to minimize the distance between two intensity distributions. We compare the distributions of the output denoised 3D gloms with our synthetic 3D blobs to make the BlobCUT have good generation quality. After obtaining the translated image ($G(I_{n})$), the number of glom can be determined by using connected component analysis, assuming there is no noise in the image. Subsequently, a random image with an equivalent number of gloms is generated using the 3D elliptical Gaussian function. These two images are then fed back into the encoder $G_{enc}$ of the generator *G* to obtain the intensity distribution representations of the images. The Kullback-Leibler Divergence (KLD) loss is calculated for the two intensity distributions. By minimizing the KLD loss, the translated image can be ensured to have a similar distribution to the synthetic image. This approach effectively addresses the issue of small object blurring that is present in the original CUT model-generated image by utilizing the intensity distribution information.
(14)$$\begin{array}{cc} L_{KLD}\left(G_{enc},S\right)= & E_{x}\left[\log p\left(G_{enc}\left(G\left(I_{n}\right)\right)\right)-\log p\left(G\left(S\right)\right)\right] \end{array}$$
where S(x) is the distribution of synthetic clean blobs. $G_{enc}$ is the encoder part of the generator.

**Final objective.** Our final objective is as follows. The generated image should exhibit realism, with a simultaneous requirement for the establishment of correspondence between patches in the input and output images. The final objective loss function becomes as follows:(15)$$\begin{array}{cc} L_{total}= & L_{GAN}\left(G,D\right)\\ \; & +\lambda_{I_{n}}L_{PatchNCE}\left(G,F,I_{n}\right)\\ \; & +\lambda_{I_{c}}L_{PatchNCE}\left(G,F,I_{c}\right)\\ \; & +\lambda_{conv}L_{convex}\left(G,HI,M\right)\\ \; & +\lambda_{KLD}L_{KLD}\left(G_{enc},S\right) \end{array}$$
where $\lambda_{I_{n}}$, $\lambda_{I_{c}}$, $\lambda_{conv}$, and $\lambda_{KLD}$ are the parameters to tune our proposed method to get good denoising performance. Once trained, BlobCUT has the ability to perform image translation for the purpose of blob image denoising and obtaining a glom/blob probability mask.

### 3.3. 3D Blob Identification through Voxel Union Constraint Operation

The BlobCUT approach presented in Section 3.2 provides the Hessian convexity mask $HI(I_{n})$ and the blob probability mask $M(G\left(I_{n}\right))$. Prior works [14] have shown that Hessian analysis tends to over-segment the blobs and the U-Net probability map tends to under-segment the blobs. To resolve the issues associated with Hessian analysis and U-Net in [14], we similarly use a voxel union constraint operation to derive the final blobs. Towards this, the final blob identification mask $M_{final}\left(I_{n}\right)$ is derived by applying the joint operation on the two masks as follows:(16)$$M_{final}\left(I_{n}\right)=HI\left(I_{n}\right)⊙M\left(G\left(I_{n}\right)\right)$$
where the joint operator ⊙ is the Hadamard product. Figure 3 shows the step-by-step methodology of blob identification through BlobCUT and joint constraint operation. Figure 3a–c shows the original noisy blobs image, the clean blob image of blobs without noise, and the centroids of the blobs’ ground truth in respective order. While Figure 3d–e shows the Hessian convexity mask and the blob mask that were generated using BlobCUT. Finally, Figure 3f shows the final blob identification mask generated by the joint operation. The blue circle in parts (a), (b), and (c) shows a total of four blobs, three of which are overlapping while the fourth is separated. In the Hessian convexity mask shown in part (d), they are correctly identified as four distinct blobs. However, In the blob mask shown in part (e), all four blobs are incorrectly segmented as one blob highlighting the under-segmentation issue discussed previously. The yellow-colored ellipse shows a segment of the noise that needs to be cleaned from the image. It is obvious from part (e) that the blob mask is able to distinguish this segment as pure noise and successfully remove it while the Hessian mask in part (d) incorrectly identifies the noise as blobs, causing some of the noisy regions to be detected as blobs. Which highlights the over-segmentation issue in the Hessian mask. Therefore, by taking the intersection of the Hessian convexity mask and the blob mask, their respective issues of under-segmentation and over-segmentation are alleviated. The effectiveness of this operation has been validated by multiple significant experiments in [32]. The final blob mask derived by the joint constraint operation is shown in Figure 3f. Clearly, it is able to selectively segment the blobs while avoiding over-segmentation of noisy regions.

We define the true 3D blob candidate as a 26-connected voxel, so the final identified blobs set are represented as:(17)$$S_{blob}=\left\{\left(i,j,k\right)|\left(i,j,k\right)\in I_{d},I_{b}\left(i,j,k\right)=1\right\}$$

## 4. Experiments and Results

### 4.1. Networks and Hardware

BlobCUT is directly trained on 3D images. We take 3D images with 64×64×32 voxels as input, then resize them to 128×128×64 voxels and normalize to [0,1]. The generator *G* adopts an encoder-decoder structure with residual blocks, similar to [50]. The Generator consists of a 7×7×7 3D Convolution-InstanceNormalization-ReLU layer. Then followed by 2 Downsampling layers, 6 Residual blocks, 2 Upsampling layers, and Tanh loss function. Each Downsampling layer is a 3×3×3 3D Convolution-InstanceNormalization-ReLU layer with stride size 2. Each Residual block consists of two 3×3×3 3D Convolution-InstanceNormalization-ReLU layer. Each Upsampling layer is a 3×3×3 3D fractional-strided-Convolution-InstanceNormalization-ReLU layer with a stride size of 2. A 3D replication padding with a padding size of 1 is used in all Convolution layers. The discriminator *D* adopts a 70×70 PatchGAN [51] for classifying the real images and translated images. It consists of three 4×4×4 3D Convolution-InstanceNormalization-LeakyReLU layers with a stride size of 2, one 4×4×4 3D Convolution-InstanceNormalization-LeakyReLU layer with a stride size of 1 and one 4×4×4 3D Convolution layer with a stride size of 1. The LeakyReLU in each layer has a slope of 0.2. Finally, a Sigmoid activation function is applied in the output layer.

Our experiments are conducted on the platform with 1 Intel CORE i7-10700K CPU, manufactured by Intel Corporation based in Santa Clara, California, USA, and 1 NVIDIA RTX 3090 GPU with 24 GB of memory, sourced from NVIDIA Corporation, also headquartered in Santa Clara, California, USA.

### 4.2. Dataset Description

We conducted two experiments to validate the performance of BlobCUT. The first experiment is applying the BlobCUT to synthetic noisy 3D blob images to see the significant improvement compared to the other methods. The second experiment is applying the BlobCUT to real mouse kidney CFE-MR images to check the segmentation performance on a real dataset. BlobCUT needs training datasets from source domain *S* and target domain *T* as input. Target domain *T* in BlobCUT should be fixed as synthetic clean 3D blobs, and all of them were generated by the function in Section 3.

In the context of the target domain, we randomly generated a dataset consisting of 1000 3D blob images, each with dimensions of 64×64×32 voxels, for use in both experimental settings. For each 3D blob image, blobs were spread in random locations. The number of blobs for each image ranged from 500 to 800. The parameters of the 3D elliptical Gaussian function for each synthetic blob are as follows: ϕ and θ are set to x $\sigma_{1}$$\sigma_{2}$ are all set to [0.5,1.5]. We record the blob mask of each image.

For source domain *S*, in the first experiments, we synthesized another 1000 3D blobs images using the same 3D blobs synthesis function in Section 3, and record their label mask as ground truth. Then random noise was added to these images to make them become noisy 3D blob images. Noise was generated by the Gaussian function with μ=0 and δ defined by:(18)$$\sigma_{noise}^{2}=\frac{\sigma_{image}^{2}}{10^{\frac{SNR}{10}}}$$
where the signal-to-noise ratio SNR was randomly set from 0.01 db to 1 db. The 1000 synthetic clean 3D blobs images and 1000 noisy 3D blobs images are treated as the training dataset for the first experiment. Figure 4 demonstrates the source domain and target domain images used by BlobCUT during training in Experiment I.

In the second experiment, we studied 62 mice kidney CFE-MR images as source domain *S*. Each mouse kidney MR image has voxel dimensions of 256×256×256. The dataset comprises two distinct comparison groups: chronic kidney disease (CKD) with a sample size of 26 individuals in contrast to 18 controls, and acute kidney injury (AKI) with 10 individuals compared to 8 controls. The animal experiments were approved by the Institutional Animal Care and Use Committee (IACUC) under protocol #3929 on 4 July 2020 at the University of Virginia, in accordance with the National Institutes of Health Guide for the Care and Use of Laboratory Animals. They were imaged by CFE-MRI as described in [52]. For validation, we split the 62 mice kidneys into training and validation sets. The training dataset includes 44 mice kidneys, and the validation dataset consists of 18 mice kidneys. We randomly sampled 1000 3D patch images (64×64×32 voxels) from 44 mice kidneys. These 3D patch images (Figure 5) have varied locations in mice kidneys and were non-overlapping. The sampling process is performed in the cortex region because the medulla region of mouse kidney does not have glomeruli. The medulla and cortex regions of the mice kidneys were meticulously annotated by an expert in the field. The training dataset for the second experiment comprises 1000 synthetic 3D blob images and 1,000 3D patch images extracted from mice kidneys. Figure 4 demonstrates the source domain and target domain images used by BlobCUT during training in Experiment II.

### 4.3. Experiment I: 3D Synthetic Image Data

Because medical kidney images with experts’ labeling are rare and expensive, we use synthetic data that have label masks (ground truth) instead. Thus, with the label masks, we can quantify the segmentation performance of different methods. Our proposed method, BlobCUT, underwent a comparative analysis against six other methods: UHDoG [13], BTCAS [14], BlobDetGAN [18], UVCGAN [16], EGSDE [17], and CUT [15]. The selection of UHDoG and BTCAS was motivated by their pretraining on a public cellular dataset, allowing us to illustrate the advantages of eliminating the reliance on additional domain information. While BlobDetGAN is a state-of-the-art method in glom segmentation, its extended training time presented a challenge. The inclusion of UVCGAN, EGSDE, and CUT (acknowledged as cutting-edge techniques in unpaired image-to-image translation) serves the purpose of highlighting the intrinsic challenges associated with small blob detection. The objective is to demonstrate that, even among the most advanced methods in unpaired image-to-image translation (UI2I), achieving satisfactory performance in addressing the small blob detection problem is challenging without specialized design considerations. This rationale informs the choice of these specific methods for comparison against our proposed approach.

We evaluated the performance of these six methods by the following seven metrics: Detection Error Rate (DER), Precision, Recall, F-score, Dice coefficient, Intersection over Union (IoU) and Blobness. For detection, DER measures the difference ratio between number of detected blobs and ground truth. Precision quantifies the proportion of retrieved candidates that are validated by the ground truth, while recall quantifies the proportion of ground-truth data that is successfully retrieved. The F-score provides a comprehensive evaluation of the combined performance of precision and recall. In the context of segmentation, the Dice coefficient assesses the similarity between the segmented blob mask and the ground truth, and the Intersection over Union (IoU) measures the degree of overlap between the segmented blob mask and the ground truth. For synthesis, Blobness evaluates the likelihood of an object having a blob-like shape.

Since the blob locations (the coordinates of the blob centers) were already generated when synthesizing the 3D blob images, the ground-truth number of blobs for all 3D images was already recorded. DER can be calculated by Equation (Equation 19).
(19)$$DER=\frac{|N_{GT}-N_{Det}|}{N_{GT}}$$
where $N_{GT}$ represents the number of ground-truth blobs and $N_{Det}$ represents the number of detected blobs. A candidate was classified as a true positive if the centroid of its magnitude fell within a detection pair (i,j), where the nearest ground truth center *j* had not been previously associated, and the Euclidean distance $D_{ij}$ between ground truth center *j* and blob candidate *i* was less than or equal to the specified threshold *d*. To avoid double-counting, the number of true positives TP was computed using Equation (Equation 20). Precision, recall, and F-score were computed as per Equations (Equation 21)–(Equation 23).
(20)$$\begin{array}{c} TP=\min\{\#\left\{\left(p_{i},p_{j}\right):\min_{i=1}^{m}D_{p_{i}p_{j}}\leq d_{\delta }\right\},\\ \#\{\left(p_{i},p_{j}\right):\min_{j=1}^{n}D_{p_{i}p_{j}}\leq d_{\delta }\}\} \end{array}$$
(21)$$Precision=\frac{TP}{n}$$
(22)$$Recall=\frac{TP}{m}$$
(23)$$F-score=2\times \frac{Precision\times Recall}{Pecision+Recall}$$
where *m* is the number of true glomeruli and *n* is the number of blob candidates; *d* represents a threshold with a positive value in the range of (0,+∞). When *d* is set to a smaller value, fewer blob candidates are considered, as it requires a closer proximity between the centroid of a blob candidate and the corresponding ground truth. Conversely, when *d* is set too large, more blob candidates are taken into account. In this context, given the potential presence of local intensity maxima within small, irregularly shaped blobs, we configure *d* to be equal to the average diameter of the blobs. The Dice coefficient and IoU were computed by comparing the segmented blob mask with the ground truth mask using Equations (Equation 24) and (Equation 25).
(24)$$Dice\left(BM,BG\right)=\frac{2|BM\cap BG|}{\left|BM|+|BG\right|}$$
(25)$$IoU\left(BM,BG\right)=\frac{BM\cap BG}{BM\cup BG}$$
where BM is the binary mask for segmentation result and BG is the binary mask for the ground truth. Based on identified blobs set $S_{blob}$, blobness for each blob candidate $b_{i}$ is calculated by Equation (Equation 26):(26)$$Blobness_{b_{i}\in S_{blob}}=\frac{3\times {\left|det\left(H\left(J-f\right)\right)\right|}^{\frac{2}{3}}}{pm(H(J-f))}$$
where *f* denotes the normalized 3D blobs image, *J* represents the unit matrix with all ones, *H* symbolizes the Hessian matrix, and pm indicates the principal minors of the Hessian matrix. We make the assumption that the blobs in *f* correspond to dark blobs in Equation (Equation 26) for consistency with other equations. It’s important to note that in [22], the calculation of blobness is based on the DoG transformed blobs image, whereas here, we use the normalized 3D blobs image without the DoG transformation. Consequently, the transformation of these dark blobs into bright blobs is achieved by J−f.

Note that since we already know blob mask of each synthetic image, we calculated the average blobness of all images as ground truth (in this experiment, the average blobness for all images is 0.519). UHDoG, BTCAS, UVCGAN and EGSDe firstly detect the gloms in 2D-space and then stack the results as 3D outputs. We limited our training time comparison to BlobCUT, BlobDetGAN, and 3D CUT, as these networks are specifically tailored for 3D segmentation, providing a more relevant and comparable benchmark.

Table 1 shows the comparative performance results of BlobCUT against SOTA methods on the synthetic data. In terms of evaluation metrics, BlobCUT attains best values in Dice, IoU, and Blobness. It demonstrates comparable performance to BlobDetGAN in DER, recall, and F-score, with a marginally lower precision. The diminished performance of UHDoG, BTCAS, UVCGAN, and EGSDE in Dice, IoU, and Blobness, relative to BlobCUT and BlobDetGAN, is unsurprising, given their reliance on two-dimensional spatial designed methods. In the assessment of F-score, the three generic methods in unpaired image-to-image translation (UVCGAN, EGSDE, and CUT) exhibit lower values than the blob detectors (UHDoG, BTCAS, BlobDetGAN, and BlobCUT). This underscores the inherent challenge of small blob detection, where generic approaches, lacking meticulous design modifications beyond alterations to the training set, struggle to attain commendable performance. Not only do the empirical findings presented in Table 1 indicate the quantitative superiority of BlobCUT against SOTA methods, but they also show that BlobCUT proves to be a faster and more efficient method for training models, completing the training process in 1393 s per epoch, a notable improvement over BlobDetGAN’s 2171 s per epoch.

Furthermore, as Figure 6 shows, it is noteworthy that BlobCUT attains its peak performance at 23 epochs, whereas BlobDetGAN reaches its optimal performance at 26 epochs. Accounting for the temporal investment per epoch, BlobCUT outperforms the state-of-the-art method BlobDetGAN, achieving superior results with only 56.6% of its training time. This emphasizes the efficacy of BlobCUT in precisely segmenting small blobs, concurrently achieving significant improvements in training efficiency.

### 4.4. Experiment II: 3D MR Images of Mouse Kidney

We conducted experiments involving CF-labeled glomeruli within a dataset of 3D magnetic resonance (MR) images to quantify the total count, denoted as $N_{glom}$, of glomeruli in both healthy and diseased mouse kidneys. The segmentation of glomeruli was performed using four blob detectors: UH-DoG, BTCAS, BlobDetGAN, and the proposed BlobCUT. Additionally, we included two advanced unpaired image-to-image (UI2I) methods, UVCGAN and EGSDE, for comparative analysis. The results of HDoG with VBGMM from [19], confirmed by physicians are used as the ground truth in this paper. The parameter settings of DoG were: window size N=7, γ=2, Δs=0.001. To denoise the 3D blob images by using trained U-Net, we firstly resized each slice to 512×512 by bilinear interpolation and each slice was fed into U-Net. The final probability map of whole kidney is reconstructed by combining all 2D patches and used for UH-DoG. The threshold for the U-Net probability map in UH-DoG was 0.5. Since mouse kidney has voxel dimensions of 256×256×256, to validate through BlobDetGAN and BlobCUT, we divided each mouse kidney as 128 3D patches (64×64×32). The final identification mask of the whole kidney is also reconstructed by combining all 3D patches. To train the BlobDetGAN, $\lambda_{cycle}$ and $\lambda_{convex}$ were set to 10, $\lambda_{identity}$ was set to 0.5. They were both trained from scratch with a learning rate 0.0002. The training typically took about 20 epochs to converge so we did not set up the decay policy for learning rate. We used the Adam optimizer with a batch size set to 1.

We perform quality control by visually checking the identified glomeruli, visible as black spots in the images. For illustration, Figure 7 shows the comparison of glomerular segmentation results on mouse kidneys of CKD group and CKD control group using UH-DoG, BTCAS, BlobDetGAN, UVCGAN, EGSDE and BlobCUT, respectively. From the zoom-in regions from Figure 7, BlobCUT and BlobDetGAN exhibit relatively superior performance, manifested in their ability to accurately identify gloms without excessive omission of correct gloms. Moreover, they demonstrate outstanding resistance to image defects, such as dark edges (#429-111, #466-096). In contrast, UHDoG and BTCAS show comparatively inferior performance. Due to its reliance on a single threshold, UHDoG’s performance is less stable, as observed in its excellent performance in #466-096 but less impressive performance in #429-111. It is prone to noise interference, resulting in an over-segmentation problem. Conversely, BTCAS exhibits more stability in performance and is less susceptible to noise interference compared to UHDoG. However, in comparison to BlobCUT, it has a lower recognition rate, attributed to the limitations imposed by utilizing a pre-trained U-Net. As for the remaining two generic models, UVCGAN and EGSDE, designed for UI2I tasks, manifest similar issues of under-segmentation and sensitivity to noise. This is attributed to their lack of dedicated geometric and intensity constraints for glom detection, which causes shape distortion and unseparated overlapping gloms during inference.

$N_{glom}$ is reported in Table 2, where the UH-DoG, BTCAS, BlobDetGAN, UVCGAN, EGSDE and the proposed BlobCUT blob detectors are compared to HDoG with VBGMM from [22]. The differences between the results are also listed in Table 2. We observe that the performance of UHDoG is not very stable, showing differences of 1% and 6% within the same experimental group, primarily attributed to its lack of robustness when employing a single threshold on the probability map generated by U-Net in conjunction with the Hessian convexity map. In contrast, BTCAS, due to its multi-threshold design, exhibits greater stability compared to UHDoG but does show significant deviations in certain kidneys (ID 427, 462, 476). As mentioned earlier, UVCGAN and EGSDE share a common issue of under-segmentation, as evident in the table. Regarding BlobCUT and BlobDetGAN, both perform well on the dataset of mouse kidney magnetic resonance images. Similar to the previous experiment, BlobCUT demonstrates yet again its computational superiority over BlobDetGAN, as it achieves this performance using approximately 60% of the training time required by BlobDetGAN.

## 5. Discussion: Denoising and Applications

### 5.1. Denoising

Imaging biomarkers derived from blob images hold substantial potential to inform clinical decision-making. Illustratively, the glomerulus-based imaging biomarkers ($N_{glom}$) examined in Experiment II stand as pivotal indicators in clinical trials, mitigating the financial and procedural burdens associated with early detection research in kidney diseases. Despite the clinical significance of these biomarkers, their historical acquisition has predominantly relied on invasive stereological methodologies, often conducted post-mortem. Notably, beyond the methodologies elucidated in this paper, a majority of AI approaches are founded upon histological section images, enhanced resolution but with higher costs.

The advanced development of Magnetic Resonance Imaging (MRI) technology facilitates the measurement of these biomarkers in a non-invasive imaging approach. However, glomeruli are relatively small compared to the imaging resolution, and they exhibit a visual appearance similar to noise. Therefore, denoising is indispensable for this kind of medical imaging detection and segmentation task.

Research [13] indicates that Hessian analysis lacks robustness against noise, making noise easily misidentified as false positives. To address this issue, UH-DoG and BTCAS utilize probability maps from a pretrained U-Net to denoise blob images. However, this may result in suboptimal denoising performance, as objects in pre trained public datasets still have different distributions from noise. To illustrate this point, Figure 8 compares the denoised results of U-Net, 3D CUT, and BlobCUT. From Figure 8b, it is evident that the blue circles in Figure 8a represent noise, but they appear as blobs in Figure 8c and are not visible in Figure 8d,e. This suggests that 3D CUT and BlobCUT outperform U-Net in denoising performance. However, noise may indirectly impact the translation of blobs, altering their shapes. Comparing the yellow circles in Figure 8b,d, it is clear that there are differences in the shapes of the blobs. The blobs within the yellow circle in Figure 8e exhibit a much similar shape to those in Figure 8b. This demonstrates that BlobCUT is capable of preserving the geometric properties of blobs during translation. However, without constraint on translation, CUT may experience geometric distortion. In conclusion, BlobCUT excels in denoising performance and effectively shields blobs from the influence of noise during the denoising process.

### 5.2. Applications

Aside from its application in detecting glomeruli in kidney MRI images, with its denoising capabilities and proficiency in detecting small blobs, BlobCUT can extend its utility to various projects in the medical imaging domain, including tasks such as nuclei/cell detection [53,54] and microparticle picking in electron microscopy imaging [55,56]. Taking nuclei detection as an illustrative example: 1. BlobCUT facilitates the identification and delineation of cell nuclei, thereby supporting the diagnosis and grading of diseases. In cancer diagnosis, for instance, the characterization of nuclei offers insights into tumor stage and aggressiveness. 2. BlobCUT enables quantitative analysis of diverse cellular features, including size, shape, and density. These objective measures contribute to a standardized assessment of tissue characteristics. 3. BlobCUT aids in comprehending the spatial distribution of cell nuclei, assisting in the development of targeted therapies and determining the extent of surgical interventions.

Moreover, It is noteworthy that the potential of BlobCUT is not limited to medical applications. BlobCUT demonstrates considerable potential across various real-world applications such as microbiological detection [57,58] and crowd counting [59,60]. One noteworthy application is in sonar image analysis, specifically for discerning objects or structures on the seafloor. Sonar images, with their diverse seabed structures, present challenges for accurate detection [61]. By integrating BlobCUT, a segmentation technique that distinguishes object features from the background, sonar images can be effectively segmented. This integration facilitates the identification of prominent objects, such as sunken ships or archaeological sites, enhancing feature detection and ensuring robustness in different seabed environments. Beyond maritime applications, BlobCUT finds relevance in Industrial Inspection and Quality Control. In industrial settings, where imagery analysis is pivotal for quality control and defect detection [62,63,64], BlobCUT can be employed to identify and segment-specific defects or irregularities in manufactured products, thereby enhancing the precision and efficiency of quality assessment processes. Additionally, BlobCUT extends its utility to Material Science and Microstructure Analysis. In the microscopic examination of materials, which often reveals intricate microstructures and particulate systems [65,66,67], BlobCUT can be valuable for segmenting and characterizing these microstructural elements. This capability facilitates advanced material analysis and supports research in fields such as metallurgy, nanotechnology, and material science.

## 6. Conclusions and Future Work

In this research, we propose BlobCUT for small blob detection and segmentation. This work provides three main contributions to the literature. First, we propose a novel 3D small blob detector BlobCUT which combines contrastive learning and generative adversarial network; Second, a blob distribution consistency constraint based on Kullback–Leibler divergence to maintain the same blobs’ distribution before and after denoising is constructed and implemented. Third, a convexity consistency constraint to preserve the blobs’ geometry property is implemented to improve the performance. In the validation experiments, BlobCUT significantly outperforms the widely used methods HDoG, UH-DoG, BTCAS and the original CUT model, yielding more accurate imaging biomarkers. Furthermore, BlobCUT exhibits comparable performance to the state-of-the-art method, requiring only around 56.6% of the training time.

While BlobCUT exhibits noteworthy efficacy in detecting small blobs, there remains scope for enhancement. The algorithm relies on a geometric assumption (e.g., for glomeruli detection is the elliptical shape) regarding blob shape, a constraint that may prove overly stringent in certain scenarios. Notably, BlobCUT exhibited a tendency to over-detect gloms in diseased kidneys of female animals, as these kidney MRIs often featured irregularly shaped gloms compared to their healthy counterparts. Furthermore, the algorithm’s performance faces skepticism from physicians due to its reliance on synthetic training data. Despite achieving high metrics in isolation, the lack of human guidance renders the results less meaningful. To address these limitations, we think it is necessary to leverage the expertise of human annotators and to optimize resource utilization using active learning based approaches. Our future endeavors include integrating human-in-the-loop active learning, a strategy designed to refine the training process. By doing so, we aim to cultivate a model that garners greater credibility among physicians and enhances the segmentation of small blobs in medical images.

## Figures and Tables

**Figure 1 bioengineering-10-01372-f001:**
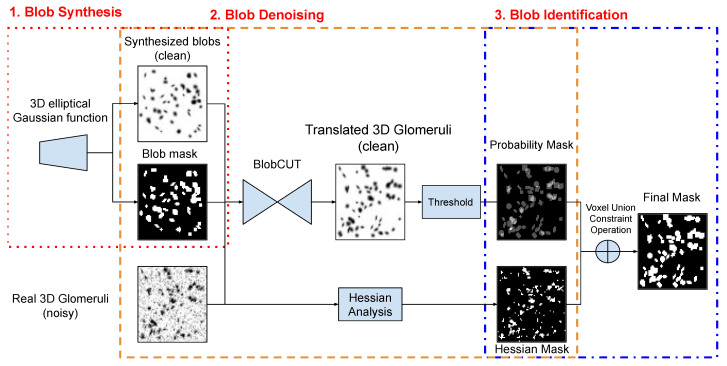
Steps to identify blobs by using BlobCUT.

**Figure 2 bioengineering-10-01372-f002:**
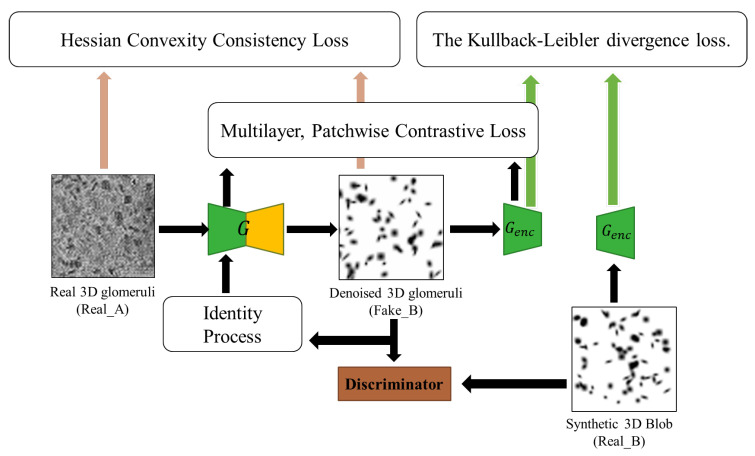
The training process of our proposed BlobCUT.

**Figure 3 bioengineering-10-01372-f003:**
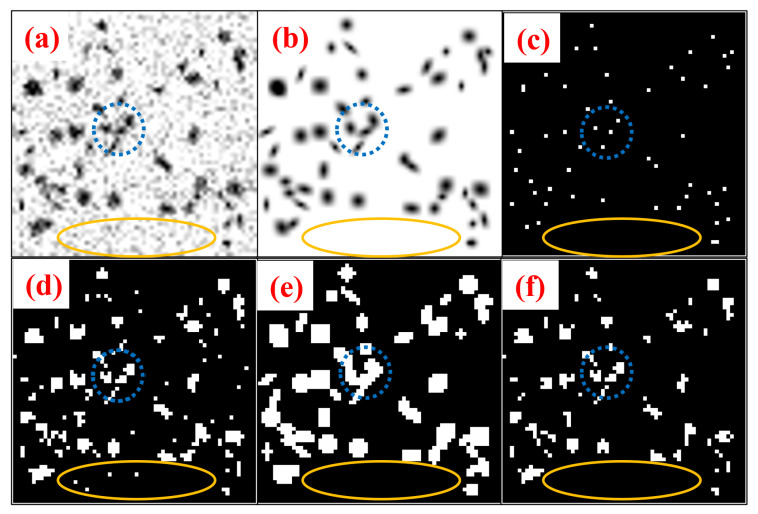
Illustration of blob identification through joint constraint operation. (**a**) Original noisy blobs image. (**b**) Clean blob image of blobs without noise. (**c**) Ground truth of blob centers. (**d**) Hessian convexity mask. (**e**) Blob mask from networks. (**f**) Final blob identification mask.

**Figure 4 bioengineering-10-01372-f004:**
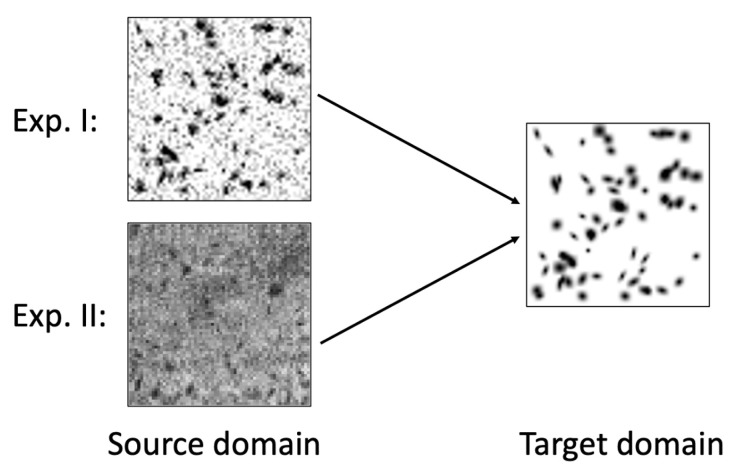
Illustration of the training datasets used by different experiments. For Exp. I, synthetic noisy blob images were used as the source domain images; for Exp. II, real kidney MR images were used as the source domain images. For both experiments, synthetic clean blob images were used as target domain images to encourage the model to have denoising capability.

**Figure 5 bioengineering-10-01372-f005:**
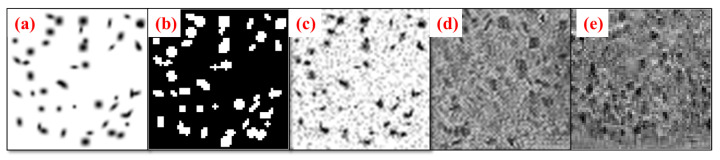
Illustration of training input images of BlobCUT (**a**) Synthesized 3D blobs image from domain clean 3D blobs. (**b**) Blob mask of (**a**). (**c**) Synthesized 3D noisy blobs image from domain noisy 3D blobs. (**d**) Synthesized 3D mouse kidney image patch from domain noisy 3D blobs. (**e**) Real 3D mice kidney image patch from domain T.

**Figure 6 bioengineering-10-01372-f006:**
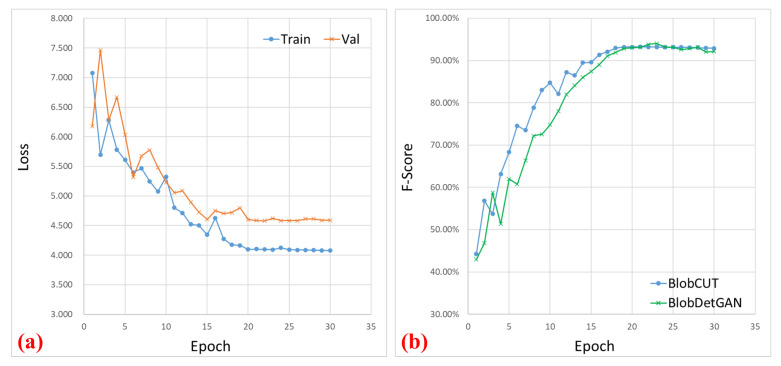
Illustration of the learning curve. (**a**) Loss curve comparison between training and validating of BlobCUT. (**b**) Testing F-score comparison between BlobCUT and BlobDetGAN.

**Figure 7 bioengineering-10-01372-f007:**
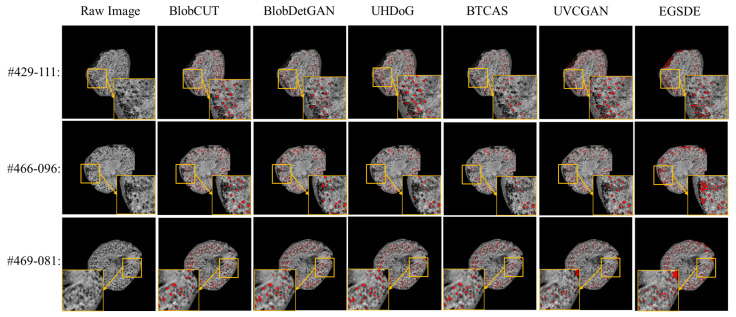
Comparison of glomerular segmentation results from 3D MR images of mouse kidneys using BlobCUT, BlobDetGAN, UH-DOG, BTCAS, UVCGAN and EGSDE. Identified glomeruli are marked in red. Three slices are illustrated: kidney #429 slice 111, #466 slice 96 and #469 slice 81.

**Figure 8 bioengineering-10-01372-f008:**
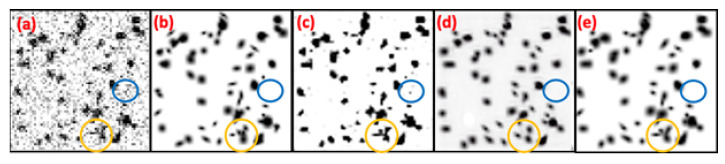
Denoising results of noisy synthetic blobs image using U-Net, 3D CUT, BlobCUT and compared with ground truth. (**a**) Original noisy blobs image. (**b**) Ground truth. (**c**) Denoised result of U-Net. (**d**) Denoised result of 3D CUT. (**e**) Denoised result of BlobCUT.

**Table 1 bioengineering-10-01372-t001:** Comparison (AVG ± STD) of UHDoG, BTCAS, UVCGAN, EGSDE, BlobDetGAN, CUT, and BlobCUT on 3D synthetic images (* significance p<0.05).

Metrics	BlobCUT	UHDoG	BTCAS	UVCGAN	EGSDE	BlobDetGAN	3D CUT
* **DER** *	**0.085 ± 0.035**	0.166±0.040	0.093±0.061	0.117±0.075	0.128±0.058	0.081±0.039	0.126±0.052
		(* < 0.001)	(* < 0.001)	(* < 0.001)	(* < 0.001)	(0.096)	(* < 0.001)
* **Precision** *	**0.972 ± 0.0013**	0.940±0.040	0.802±0.086	0.867±0.026	0.901±0.018	0.980±0.013	0.954±0.015
		(* < 0.001)	(* < 0.001)	(* < 0.001)	(* < 0.001)	(* < 0.001)	(* < 0.001)
* **Recall** *	**0.896 ± 0.031**	0.786±0.029	0.988±0.004	0.825±0.016	0.812±0.028	0.904±0.034	0.736±0.052
		(* < 0.001)	(* < 0.001)	(* < 0.001)	(* < 0.001)	(0.786)	(* < 0.001)
* **F-score** *	**0.932 ± 0.018**	0.856±0.026	0.885±0.072	0.845±0.021	0.854±0.026	0.940±0.039	0.830±0.021
		(* < 0.001)	(* < 0.001)	(* < 0.001)	(* < 0.001)	(0.062)	(* < 0.001)
* **Dice** *	**0.816 ± 0.061**	0.296±0.005	0.348±0.012	0.475±0.011	0.458±0.009	0.795±0.005	0.587±0.020
		(* < 0.001)	(* < 0.001)	(* < 0.001)	(* < 0.001)	(* < 0.001)	(* < 0.001)
* **IoU** *	**0.694 ± 0.074**	0.174±0.003	0.210±0.009	0.462±0.013	0.429±0.009	0.666±0.004	0.548±0.023
		(* < 0.001)	(* < 0.001)	(* < 0.001)	(* < 0.001)	(* < 0.001)	(* < 0.001)
* **Blobness** *	**0.542 ± 0.291**	0.550±0.301	0.562±0.300	0.560±0.321	0.563±0.315	0.548±0.213	0.551±0.299
(GT:0.519)		(* < 0.001)	(* < 0.001)	(* < 0.001)	(* < 0.001)	(* < 0.001)	(* < 0.001)
* **Avg. Time** *	**1393**	\	\	\	\	2176	1246
(s/epoch)							

**Table 2 bioengineering-10-01372-t002:** Mouse kidney glomerular segmentation ($N_{glom}$) from CFE-MRI using UH-DoG, BTCAS, UVCGAN, EGSDE, BlobDetGAN and the proposed BlobCUT compared to HDoG with VBGMM method (difference % compared with results of HDoG with VBGMM). Results with bold font are the best.

Mouse Kidney		HDoG with VBGMM (Ground Truth)	BlobCUT (Diff. %)	BlobDet-GAN (Diff. %)	UH-DoG (Diff. %)	BTCAS (Diff. %)	UVCGAN (Diff. %)	EGSDE (Diff. %)
**CKD**	ID 429	7656	7621 (0.46%)	**7633 (0.30%)**	7346 (4.05%)	7719 (0.82%)	7358 (3.89%)	7295 (4.72%)
	ID 466	8665	**8843 (2.05%)**	8912 (2.85%)	8138 (6.08%)	8228 (5.04%)	8365 (3.46%)	8317 (4.01%)
	ID 467	8549	8812 (3.08%)	8802 (2.96%)	8663 (1.33%)	**8595 (0.54%)**	8368 (2.11%)	8128 (4.92%)
	AVG	8290	**8425 (1.63%)**	8449 (1.92%)	8049 (2.91%)	8181 (2.13%)	8030 (3.13%)	7913 (4.54%)
**Control for CKD**	ID 427	12,724	12,573 (1.18%)	12,683 (0.32%)	**12,701 (0.18%)**	12,008 (5.63%)	12,486 (1.87%)	12,423 (2.37%)
	ID 469	10,829	**10,897 (0.63%)**	10,921 (0.85%)	11,347 (4.78%)	11,048 (2.02%)	10,604 (2.08%)	10,458 (3.43%)
	ID 470	10,704	10,579 (1.17%)	**10,774 (0.65%)**	11,309 (5.65%)	10,969 (2.48%)	10,281 (3.95%)	10,299 (3.78%)
	ID 471	11,934	12,488 (4.56%)	12,692 (6.27%)	12,279 (2.81%)	**12,058 (0.96%)**	11,685 (2.08%)	11,718 (1.81%)
	ID 472	12,569	**12,590 (0.16%)**	12,786 (1.73%)	12,526 (0.34%)	13,418 (4.75%)	11,952 (4.90%)	12,152 (3.31%)
	ID 473	12,245	12,058 (1.53%)	12,058 (1.53%)	11,853 (3.20%)	**12,318 (0.60%)**	12,025 (1.79%)	11,825 (3.43%)
	AVG	11,836	**11,864 (0.24%)**	11,986 (1.27%)	12,003 (1.41%)	11,970 (3.07%)	11,505 (2.77%)	11,479 (2.99%)
**AKI**	ID 433	11,046	11256 (1.90%)	11618 (5.18%)	**11,033 (0.12%)**	10,752 (2.66%)	10,826 (1.99%)	10,751 (2.67%)
	ID 462	11,292	**11,420 (1.13%)**	11,445 (1.35%)	10,779 (4.54%)	10,646 (5.75%)	10,892 (3.54%)	11,082 (1.86%)
	ID 463	11,542	11533 (0.07%)	**11,544 (0.02%)**	10,873 (5.80%)	11,820 (2.41%)	11,058 (4.19%)	11,282 (2.25%)
	ID 464	11,906	**11,704 (1.70%)**	11,562 (2.89%)	11,340 (4.75%)	11,422 (3.33%)	11,526 (3.19%)	11,290 (5.17%)
	AVG	11,447	**11,478 (0.27%)**	11,542 (0.83%)	11,006 (3.85%)	11,015 (3.78%)	11,075 (3.24%)	11,101 (3.01%)
**Control for AKI**	ID 465	10,336	10,482 (1.41%)	10,214 (1.18%)	10,115 (2.14%)	**10,393 (0.55%)**	10,205 (1.26%)	10,059 (2.68%)
	ID 474	10,874	**10,928 (0.50%)**	10,955 (0.74%)	11,157 (2.60%)	11,034 (1.47%)	10,989 (1.05%)	10,428 (4.10%)
	ID 475	10,292	10,196 (0.91%)	**10,222 (0.68%)**	10,132 (1.55%)	9985 (2.98%)	9982 (3.01%)	9907 (3.74%)
	ID 476	10,954	10,764 (1.71%)	11452 (4.55%)	**10,892 (0.57%)**	11,567 (5.60%)	10,735 (2.00%)	10,429 (4.79%)
	ID 477	10,885	11,047 (1.49%)	**10,929 (0.40%)**	11,335 (4.13%)	11,143 (2.37%)	10,789 (0.88%)	10,216 (6.15%)
	AVG	10,668	**10,683 (0.14%)**	10,754 (0.81%)	10,726 (0.54%)	10,824 (2.59%)	10,540 (1.20%)	10,208 (4.32%)

## Data Availability

The data presented in this study are available on request from the corresponding author. The data are not publicly available due to confidentiality.

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
