# Peer review of "BlobCUT: A Contrastive Learning Method to Support Small Blob Detection in Medical Imaging"

_bioengineering, 2023, doi:10.3390/bioengineering10121372_

Round 1
Reviewer 1 Report
Comments and Suggestions for Authors
The development of BlobCUT, a 3D small blob detector, represents a significant advancement in the challenging domain of medical imaging-based biomarker detection. Using an unpaired image-to-image translation model, BlobCUT leverages a blob synthesis module to generate synthetic 3D blobs and masks for ground truth, enhancing detection accuracy. Notably, it incorporates two key constraints, preserving geometric properties and intensity distribution, while also acting as a denoising tool, improving blob identification. In rigorous testing on both simulated and real 3D datasets, BlobCUT outperformed four state-of-the-art methods, delivering superior results in various metrics and requiring significantly less training time. The idea is interesting, but the paper needs some improvements like:
1) Some results information should be added in the end of abstract
2) Why is it better solution to use your solution than the classic u-net?
3) The background should be updated to latests solutions. Discuss the current state of neural networks like: neuro-heuristic anlaysis of video, ROI analysis with neural netwroks for sonars.
4) Is it possible to use your solution in other applications? I mentioned sonar applications, where segmentation is crucial, maybe it could be your future works? If it possible add some discussion on it.
5) Expalin in more detials loss function. Why there is five different components? I did not get from text.
6) Experimental section is very nice, but some comparison with state of art would be needed. Maybe in the form of table?
Reviewer 2 Report
Comments and Suggestions for Authors
The paper presents a novel 3D method called BlobCUT for detecting and segmenting small blobs by Contrastive Unpaired Translation. The method is evaluated on a 3D simulated dataset of blobs and a 3D MRI dataset 12 of mouse kidneys. In general, the paper is well written in English and it presents relevant information to introduce the problem.
My main concerns are the following:
a) Please improve the quality of Figure 1.
b) Please present more Figures about the results
c) Please compare the proposed method with more methods of the state of the art.
d) Please update references, many references are from the same author.
e) Please present the confusion matrix of the experiments
f) The paper is based on the previous work [13,21], please clearly indicate the main contribution of the present paper.
g) In the sentence: we employ the aforementioned formulas to generate 3D blobs in the same way of [13, 21] and designate their masks as ground truth. Please explain the method in detail, a single equation is not enough.
The paper is well written in English
Reviewer 3 Report
Comments and Suggestions for Authors
This paper is about BlobCUT: A Contrastive Learning Method to Support Small Blob Detection in Medical Imaging. My comments are as follows:
1. The abstract section is incomplete. The authors should provide more information, such as the results of their methods.
2. The introduction section is poorly organized and needs to be rewritten. The authors should mention important artificial intelligence methods in this section, including their advantages and disadvantages.
3. Please add a section for literature review. In the literature review section, it would be helpful to have a tabular summary of the reviewed papers from 2021-2023. Some articles can be presented in text format, while the remaining papers can be summarized in a table.
4. Please explain more for dataset.
5. Please provide more experimental results for your proposed model, including additional figures for loss curves, learning curves, and hyperparameters.
6. It would be beneficial to include a table that presents all the hyperparameters used in your proposed model.
7. The results section need to more discuss. The authors should present more results.
8. Please provide more experimental results for your proposed model.
9. The research question(s) should be stated more clearly and with stronger emphasis.
10. Please clarify your initial hypothesis.
11. The validation method is unclear.
12. In the discussions section, critically analyze your work/results in relation to your hypothesis.
13. Clearly identify the main findings and provide justification for the novelty and contribution of your work.
14. Recap all relevant parameters with their meanings to assist the reader's understanding.
15. Introduce a section on the limitations of the study.
16. In the Conclusion section, elaborate more on future directions for research. For example, your proposed method can be used in medical applications such as medical image and signals analysis. I recommended some references include https://doi.org/10.3389/fnmol.2022.999605, https://doi.org/10.1007/s11571-022-09897-w for this section.
17. Rewrite the conclusion section by summarizing your method, findings, and comparing them with more than 20 papers accepted in 2021-2023. Include a table for better comparison.
18. While English language usage is generally acceptable, there are some errors that need to be corrected.
Comments on the Quality of English LanguageWhile English language usage is generally acceptable, there are some errors that need to be corrected.
Round 2
Reviewer 1 Report
Comments and Suggestions for Authors
The paper can be accepted
Reviewer 2 Report
Comments and Suggestions for Authors
My comments have been properly addressed.
Comments on the Quality of English LanguageThe paper is well written in English.
Reviewer 3 Report
Comments and Suggestions for Authors
This version can be accepted for publication.